# Investigation on Battery Thermal Management Based on Enhanced Heat Transfer Disturbance Structure within Mini-Channel Liquid Cooling Plate

**Renzheng Li** [1,2,*]**, Yi Yang** [3] **, Fengwei Liang** [4]**, Jichao Liu** [5,6] **and Xinbo Chen** [1,2]

1  School of Automotive Studies, Tongji University, Shanghai 201804, China
2  Clean Energy Automotive Engineering Center, Tongji University, Shanghai 201804, China
3  School of Automotive and Traffic Engineering, Jiangsu University, Zhenjiang 212013, China
4  School of Mechanical Engineering, University of Science and Technology Beijing, Beijing 100083, China
5  Jiangsu Advanced Construction Machinery Innovation Center Ltd., Xuzhou 221000, China
6  School of Materials and Physics, China University of Mining and Technology, Xuzhou 221116, China
*  Correspondence: 1911070@tongji.edu.cn

**Abstract:** The battery thermal management system plays an important role in the safe operation of a lithium-ion battery system. In this paper, a novel liquid cooling plate with mini-channels is proposed and is improved with disturbance structures. First, an accurate battery heat generation model is established and verified by experiments. The error is less than 4%, indicating the heat generation power is reliable. Then, five designs are proposed first to determine the suitable number of disturbance structures, and plan 3 with five disturbance structures shows a satisfying performance in heat dissipation and flow field. Moreover, four layout plans are proposed, namely uniform, interlaced, thinning, and gradually denser distribution. Results show that plan 5 (uniform) achieves the best performance: the maximum average temperature is 36.33 °C and the maximum average temperature difference is 0.16 °C. At last, the orthogonal experiment and range analysis are adopted to optimize the structure parameters. Results show that the best combination is space between adjacent disturbance structures d1 = 20 mm, length d2 = 5 mm, width d3 = 1.5 mm, and tilt angle $\beta = 60°$.

**Keywords:** lithium-ion battery; thermal management; liquid cooling; mini-channel; disturbance structure

## 1. Introduction

With the rapid development of transportation electrification, lithium-ion batteries are considered an ideal power source because of their high energy density, low self-discharge rate, and long life [1]. However, lithium-ion batteries have unavoidable defects and need to be operated in a suitable but narrow temperature interval [2]. The best temperature range for lithium-ion batteries is 25–40 °C [3]. During the charging and discharging process, lithium-ion batteries generate a lot of heat, the available energy will be limited, and the battery life will decrease more quickly if the temperature is beyond the range [4]. In addition, hundreds and even thousands of batteries are connected in parallel and series to form the battery systems in practical electric vehicles [5]. According to existing research and engineering experience, the temperature difference of all the batteries in the system should be kept below 5 °C to maintain a stable and safe operation [6]. Therefore, an efficient battery thermal management system (BTMS) is necessary for the battery system to maintain the temperature in the suitable range.

There are many different ways to classify the battery thermal management systems. According to whether external energy is consumed, the BTMSs can be divided into three categories, namely active, passive, and hybrid [7–9]. Another common way is based on the type of cooling medium, and the BTMSs can be divided into five categories, namely air

cooling, liquid cooling, phase change material (PCM), heat pipe, and hybrid cooling [10–14]. A structural ventilation path or a few fans are used in air cooling and the design is simple. This cooling way is widely used in the early time and small battery systems for its energy efficiency, low cost, and long-lasting reliability and durability [15]. Saw et al. [16] utilized the CFD method to analyze the air cooling of a battery pack comprising 38,120 cells. The simulation results demonstrate that an increase in the cooling airflow rate will increase the heat transfer coefficient and pressure drop. Chen et al. [17] proposed an improved air cooling design with Z-type flow and adjustable battery pacing. Results show that the new design achieved better heat dissipation. Although air cooling has many advantages, it cannot meet the increasing demand for heat dissipation due to the low thermal conductivity of air [18]. The BTMS based on PCM dissipates heat by storing the energy with the latent heat. When the temperature increases, the PCM receives heat, and the state changes. When the temperature decreases to the phase change point, heat is released again and the material returns to the original state again [19,20]. In recent years, PCM cooling has been of great interest due to its significant cooling performance. Huang et al. [21] proposed a novel composite PCM to serve BTMSs. Results show that the material can reduce the contact thermal resistance and improve the cooling performance. Luo et al. [22] proposed a new PCM consisting of paraffin with dual-phase transition ranges, expanded graphite, and epoxy resin. Results show that the PCM can achieve great cooling performance under a 4C discharging rate. However, a significant problem is that a large amount of PCM is needed to achieve the ideal heat dissipation performance, which adds too much weight to the battery system and makes the energy density drop a lot. In addition, leakage cannot be avoided when the PCM changes to the liquid state. Heat pipes essentially use the phase change of materials to dissipate heat as well [23,24]. Due to the outstanding heat transfer performance, heat pipes with PCM are commonly studied for BTMSs, and this combination provides better heat dissipation performance than the above several ways [25]. However, the complex structure, high cost, and low energy density are still inevitable problems limiting its application in electric vehicles.

Compared to the above several cooling ways, liquid cooling has many advantages, such as high heat dissipation performance, high engineering applicability, and high energy density [26]. By adding tubes or cooling plates around the batteries, the coolant is driven by pumps and flows along the paths to dissipate the heat [27,28]. In addition, another form of liquid cooling is submerging the batteries directly in the coolant, and this way can significantly improve power consumption and temperature uniformity [29,30]. However, direct liquid cooling needs complex sealing structures, and its reliability is hard to maintain in real applications. Therefore, much attention is paid to indirect liquid cooling. Zhou et al. [31] designed a cold plate with half-helical ducts and compared its performance with the jacket liquid cooling. Results show that half-helical ducts are more efficient. Rao et al. [32] proposed a novel form of liquid cooling in which the contact surface is variable. Results show that the maximum temperature of the battery can be efficiently controlled under 40 °C. Shang et al. [33] proposed a novel liquid-cooled BTMS with changing contact surface to improve the heat dissipation performance. Results show that the battery temperature is proportional to the inlet temperature. Under the requirement of the increased energy density of the battery system, more attention has been paid to liquid cooling plates with mini-channels due to their tight size, high heat transfer efficiency, and low weight [34,35]. S. Panchala et al. [36,37] developed a new CFD model to research the heat flow field of a mini-channel liquid cooling plate; they investigated the temperature and velocity distribution of liquid-cooled LiFePO4 thermal management and revealed that increasing the battery charge–discharge rate will cause the battery temperature to rise rapidly. Huang et al. [38] applied the streamlined concept to the design of the micro-channels, and they found that the streamlined structure can reduce flow resistance and strengthen heat dissipation. Liu et al. [39] researched the effect of different types of coolant on the cooling efficiency of a micro-channel liquid cooling plate, and the results showed that water achieved the best cooling effect; then, nanofluids were added to the water, and the authors found that

nanoparticle addition can relieve the temperature rise of the battery. According to the existing research, improving the flow field of the mini-channel is an effective way, but most research focuses on the channel shape design. Improving the flow state within the original channel is still an effective way. Contributions of this paper can be shown in the following aspects:

1.  Novel mini-channel design with disturbance structure: A novel mini-channel liquid cooling plate with is proposed with improved disturbance structures. High heat dissipation and temperature uniformity are achieved based on the BTMS.
2.  Efficient mini-channel structure optimization method: Orthogonal experiment design especially for the disturbance structure is proposed to improve the designing process significantly.

## 2. Battery Heating Generation Power Experiment

### 2.1. Battery Property

In this paper, a commercial 40 Ah lithium-ion battery is selected. The heating generation power of the battery is obtained based on the simulation first, and then experiments are conducted to verify the simulation results. The battery properties are provided by the manufacturer, as shown in Table 1.

**Table 1.** Battery properties.

| Content | Parameter |
| --- | --- |
| Battery type | NCM |
| Nominal capacity | 40 Ah |
| Nominal voltage | 3.6 V |
| Weight | 825 g $\pm$ 10 g |
| Size | 28 mm$\times$ 148 mm $\times$ 93 mm |
| DC internal resistance | $\leq$2.0 m$\Omega$ |
| AC internal resistance | $\leq$1.0 m$\Omega$ |
| Density | 2140 kg/m$^3$ |
| Thermal conductivity | $\lambda_x$ = 1.5 W/(m·K) <br> $\lambda_y$ = 20.6 W/(m·K) <br> $\lambda_z$ = 20.6 W/(m·K) |
| Heat capacity | 1030 J/(kg·K) |

### 2.2. Battery Model and Verification

In the charging and discharging process of lithium-ion batteries, heat is generated along with the internal electrochemical reactions. The heat can be divided into four parts, namely reaction heat, polarization heat, Joule heat, and heat of side reaction [17]. The expression is shown as Equation (1).

$$Q = Q_r + Q_p + Q_j + Q_s \tag{1}$$

where $Q_r$ means reaction heat, $Q_p$ means polarization heat, $Q_j$ means Joule heat, and $Q_z$ means the heat of side reaction. In addition, the battery temperature will not exceed 50 °C, so the reaction heat takes a very small part and side reactions will not happen under the temperature. Therefore, the heating generation expression can be simplified as Equation (2).

$$Q = Q_j + Q_p = I^2 R = I^2(R_j + R_p)t \tag{2}$$

where $R_j$ means Joule resistance, $R_p$ means polarization resistance, $t$ means reaction time, and $I$ means current.

Based on the above analysis, the Bernardi battery model is adopted in the simulation, and its expression is shown in Equation (3) [29].

$$P = I\left[(U_{oc} - U) + T\frac{\partial U_{oc}}{\partial T}\right] \tag{3}$$

where $P$ means the battery heating generation power, $U_{oc}$ means the battery open circuit voltage, $U$ means the terminal voltage, $T$ is the battery temperature, and $I$ is the charging and discharging current.

To verify the accuracy of the battery model, the temperature rise experiment was conducted. Two PT100 temperature sensors were mounted on the side faces of the battery as shown in Figure 1. The battery temperature during the discharging process at 1 C, 2 C, and 3 C is assumed as the average value of the two sensors. Figure 2 shows the simulation and experiment results; it can be seen that the temperature rising rate and the maximum temperature increase obviously with the increase in the discharging rate. In addition, the simulation temperature is a little higher than that in the experiment because the heat exchange between the battery and ambient air cannot be avoided completely in the experiment like the setup in the simulation. The maximum temperature difference between the simulation and experiment is 0.38 °C, 0.95 °C, and 2.08 °C at 1 C, 2 C, and 3 C, respectively. The error is less than 4%, indicating that the model is accurate. In addition, the heating generation power is calculated according to Equation (4) based on the experimental results, as shown in Table 2.

$$P = \frac{Q_t}{t} = \frac{cm\Delta T}{t} \tag{4}$$

where $Q_t$ means the heat generated by the battery, $c$ means the heat capacity of the battery, $m$ means battery weight, $\Delta T$ means temperature rise in the discharging process, and $t$ means the discharging time.

**Table 2.** Heating generation power of 40 Ah battery at different discharge rates.

| Discharging Rate | 1 C | 2 C | 3 C |
|---|---|---|---|
| Time (s) | 3600 | 1800 | 1200 |
| Heating generation power (W) | 3.24 | 11.48 | 22.18 |

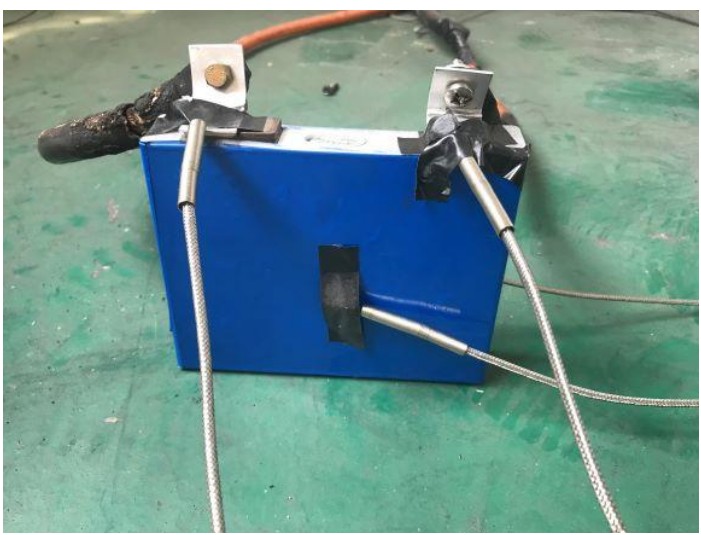

**Figure 1.** Experiment setup.

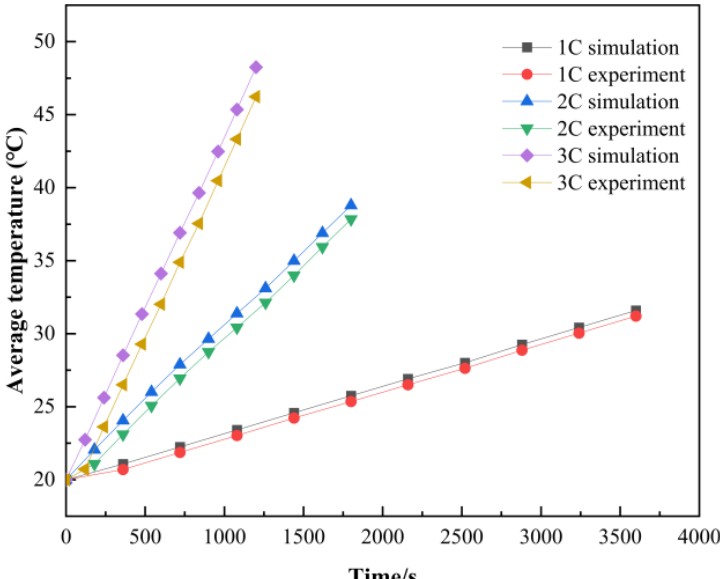

**Figure 2.** Battery model verification.

### 2.3. System Model

In practice, battery packs usually contain batteries, heat dissipation components, mechanical components, and control modules. However, the heat exchange mainly occurs between the heat dissipation components and the battery, and the influence of the remaining components is small. In order to optimize the computational performance, a battery module model consisting of 12 batteries and 6 mini-channel liquid cooling plates was built for analysis in this study, as shown in Figure 3. The batteries are numbered from 1 to 12. In addition, thermal conductive sheets are set between adjective batteries and mini-channel liquid cooling plates to decrease the contact thermal resistance. The thickness is 2 mm and the thermal conductivity is $2 \, W \times K^{-1} \times m^{-1}$. In the primary design, 2 parallel 3 tandem flow channels are used; the width is 7.6 mm and the height is 2 mm for one flow channel. In addition, the material of the plate is set as aluminum in all the simulations, and the properties are the defaults in the software. The other structural parameters are shown in Figure 3 as well.

In practical engineering applications, adding disturbance design to change the original velocity field of the fluid to make a sudden change in the flow direction and produce secondary flow, or adding fins, manifolds, and other microstructures on the flow channel to change the fluid boundary layer and flow state, can improve the heat dissipation performance without generating additional energy consumption. In this paper, the disturbance structure is applied to the design of the mini-channel liquid cooling plate, as shown in Figure 4. The structure parameters include the number of the disturbance structure within a single channel $N$, the spacing between two adjacent disturbance structures d1, the length of one disturbance structure d2, the width of one disturbance structure as well as the depth of the cavity d3, and the tilt angle of the cavity $\beta$.

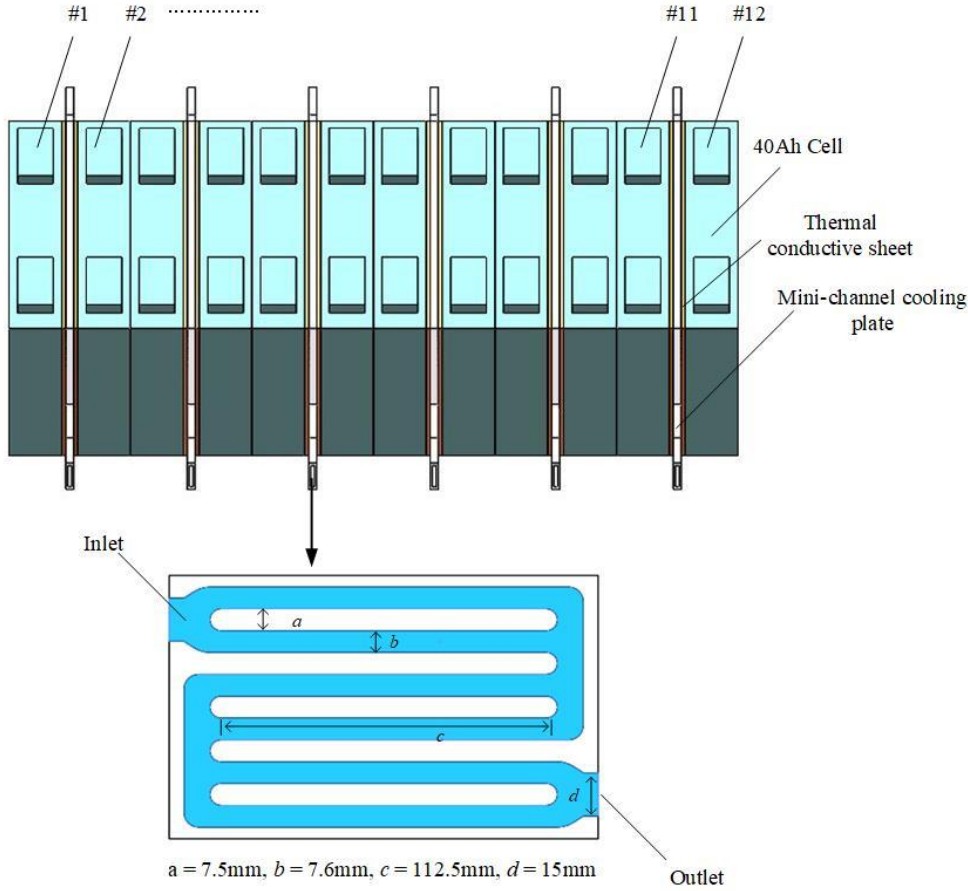

**Figure 3.** Design of the mini-channel liquid-cooling battery module.

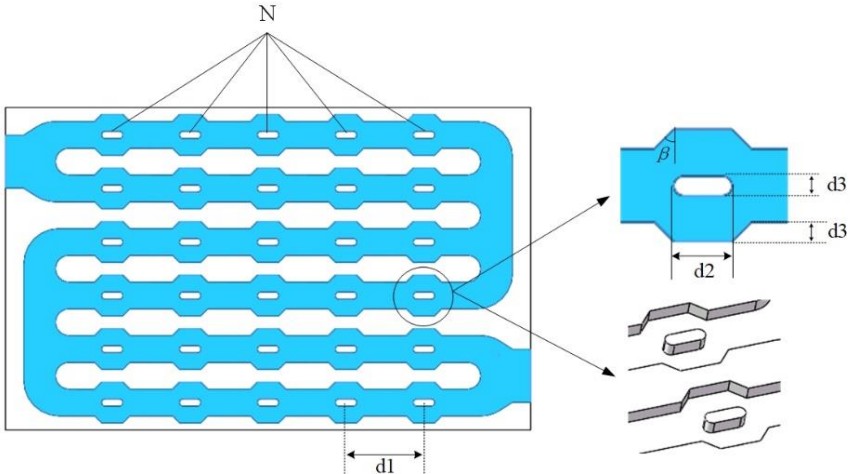

**Figure 4.** Design of the disturbance structure.

### 2.4. Computational Fluid Control Model

For the proposed mini-channel liquid cooling battery module, the flow and heat transfer process meet the conservation of mass, momentum, and energy equations [21]. The detailed expressions are as follows:

Mass conservation equation:

$$\frac{\partial \rho}{\partial t} + div(\rho u) = 0 \tag{5}$$

where $\rho$ means density, $u$ means velocity vector, and $t$ means time.



Momentum conservation equation:

$$\frac{\partial(\rho u_x)}{\partial t} + div(\rho u_x u) = div(\mu grad u_x) - \frac{\partial p}{\partial x} + R_x \tag{6}$$

$$\frac{\partial(\rho u_y)}{\partial t} + div(\rho u_y u) = div(\mu grad u_y) - \frac{\partial p}{\partial y} + R_y \tag{7}$$

$$\frac{\partial(\rho u_z)}{\partial t} + div(\rho u_z u) = div(\mu grad u_z) - \frac{\partial p}{\partial z} + R_z \tag{8}$$

where $\mu$ denotes dynamic viscosity; $u_x$, $u_y$, and $u_z$ denote the components of $u$ in $x$, $y$, and $z$ directions; $p$ denotes pressure on the computational domain; $R_x$, $R_y$, and $R_z$ denote generalized source terms.

Energy conservation equation:

$$\frac{\partial(\rho T)}{\partial t} + div(\rho u T) = div(\frac{\lambda}{C} grad T) + R_T \tag{9}$$

where $\lambda$ denotes thermal conductivity, $C$ denotes specific heat capacity, and $R_T$ denotes heat source.

Mathematical formula of battery temperature field:

$$\rho c \frac{\partial T}{\partial t} = \lambda_x \frac{\partial^2 T}{\partial x^2} + \lambda_y \frac{\partial^2 T}{\partial y^2} + \lambda_z \frac{\partial^2 T}{\partial z^2} + q \tag{10}$$

where $q$ means heat generation per unit volume of battery and $\lambda_x$, $\lambda_y$, and $\lambda_z$ denote the thermal conductivity in $x$, $y$, and $z$ directions, respectively.

Turbulence equation:

Turbulent flow phenomena exist in nature and in various engineering fields. Convective heat transfer during turbulent flow is a common method of heat transfer in engineering. Applying the standard model is currently the main method for solving the flow and heat transfer [26,27]. In this paper, the turbulence is considered according to Equation (11).

$$\rho \frac{\partial k}{\partial t} + \rho u_j \frac{\partial k}{\partial x_j} = \frac{\partial}{\partial x_j} \left[ \left( \eta + \frac{\eta_t}{\sigma_k} \right) \frac{\partial k}{\partial x_j} \right] + \eta_t \frac{\partial u_i}{\partial x_j} \left( \frac{\partial u_i}{\partial x_j} + \frac{\partial u_j}{\partial x_i} \right) - \rho \varepsilon \tag{11}$$

where $\varepsilon$ is the dissipation rate, and its control equation is shown as follows:

$$\rho \frac{\partial \varepsilon}{\partial t} + \rho u_k \frac{\partial \varepsilon}{\partial x_k} = \frac{\partial}{\partial x_k} \left[ \left( \eta + \frac{\eta_t}{\sigma_\varepsilon} \right) \frac{\partial \varepsilon}{\partial x_k} \right] + \frac{c_1}{k} \eta_t \frac{\partial u_i}{\partial x_j} \left( \frac{\partial u_i}{\partial x_j} + \frac{\partial u_j}{\partial x_i} \right) - c_2 \rho \frac{\varepsilon^2}{k} \tag{12}$$

*2.5. Boundary Condition*

The flow of the coolant fluid in the entire computational domain is set as a steady-state, steady incompressible turbulent flow; all walls are considered hydraulically smooth. K-epsilon standard turbulence equation and standard wall function are used in simulation calculation; the non-coupling implicit algorithm, second-order solution accuracy, and the energy equation are selected as well. Stop criteria are the maximum number of internal iteration steps and maximum physical time.

In addition, the heating generation power of batteries at different discharging rates has been determined by experiments. The volume flow inlet boundary condition is considered, and the value is set as 6 L/min; the pressure outlet boundary condition is considered, and the pressure value is standard atmospheric pressure. The ambient temperature is set as 27 °C. The heat exchange between the batteries and the air is considered according to Newton's law of cooling, and the expression is shown in Equation (13).

$$Q_e = hA(T_b - T_a) \tag{13}$$

where $Q_e$ means the exchanged heat, $h$ is the heat transfer coefficient, $A$ means the surface area of the batteries, $T_b$ means the battery temperature, and $T_a$ means ambient temperature.

*2.6. Grid independence Validation*

In this paper, the hexahedral mesh is used for mesh construction in the simulation. To ensure computation accuracy, mesh densification is used for the fluid area, and the basic size is 0.2 mm. Shapes of the cooling plate and thermal conductive sheets are simple and regular, which have low requirements for the density of the grid, so larger grids can be used. The basic size is 1 mm. In order to ensure the accuracy of the calculation, the grid independence validation of the numerical model is conducted. Figure 5 shows the simulation results of the highest temperature and maximum temperature difference of the battery module under a 1 C discharging rate with different grid numbers. It can be seen that when the number of grids exceeds $8.4 \times 10^5$, the calculation results tend to be stable, which means that the simulation results are not affected by the number of grids. Therefore, the mesh number of $8.4 \times 10^5$ is adopted.

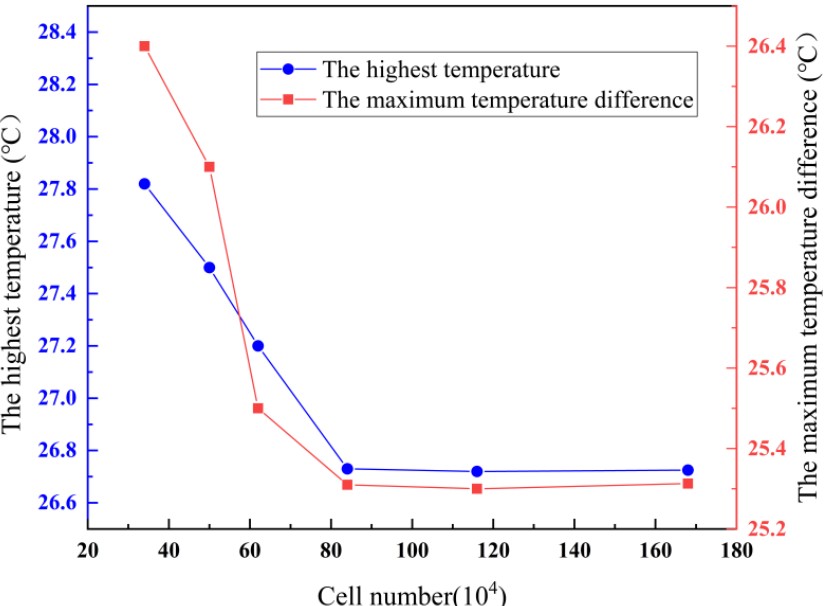

**Figure 5.** Grid dependency test.

## 3. The Influence of Disturbance Structure Arrangement on Enhanced Heat Transfer Performance

*3.1. Influence of Disturbance Structure Number*

In the liquid-cooled battery thermal management system, the coolant needs to be driven by an onboard water pump to circulate the flow, which requires additional system energy consumption and has an impact on the range of the whole vehicle. The pressure drop and friction factor in the flow channel are the main factors affecting energy consumption, and a larger pressure drop and friction factor indicate a higher pump power required The friction factor is the group of uncaused times when the fluid flows in the flow channel, and its expression is given in Equation (14).

$$f = \frac{\Delta P}{\frac{\rho u^2}{2} * \frac{L}{d_H}} \tag{14}$$

where $f$ means the friction factor; $\Delta P$ is the pressure drop of the fluid in the channel, Pa; $u$ is the fluid flow rate, m/s; $L$ is the length of the channel, m; and $d_H$ is the hydraulic diameter of the flow section, m.

According to the heat transfer theory, it is known that the increase in heat transfer area can enhance heat transfer efficiency. Adding the disturbance structure inside the mini-channel can change the flow state and increase the contact area with the coolant, which promotes the rapid heat transfer between the coolant and the mini-channel cooling plate. However, the disturbance structure also causes external pressure loss and frictional resistance. In order to study the influence of disturbance structure number, five design plans are compared, as shown in Figure 6. The plans include one, three, five, and seven disturbance structures. The original plan has none. The structure parameters of the disturbance in different plans are all length (mm) × width (mm) × height (mm) × tilt angle = 4 × 2 × 2 × 45°. In addition, the simulation conditions are all set as discharging at 3 C.

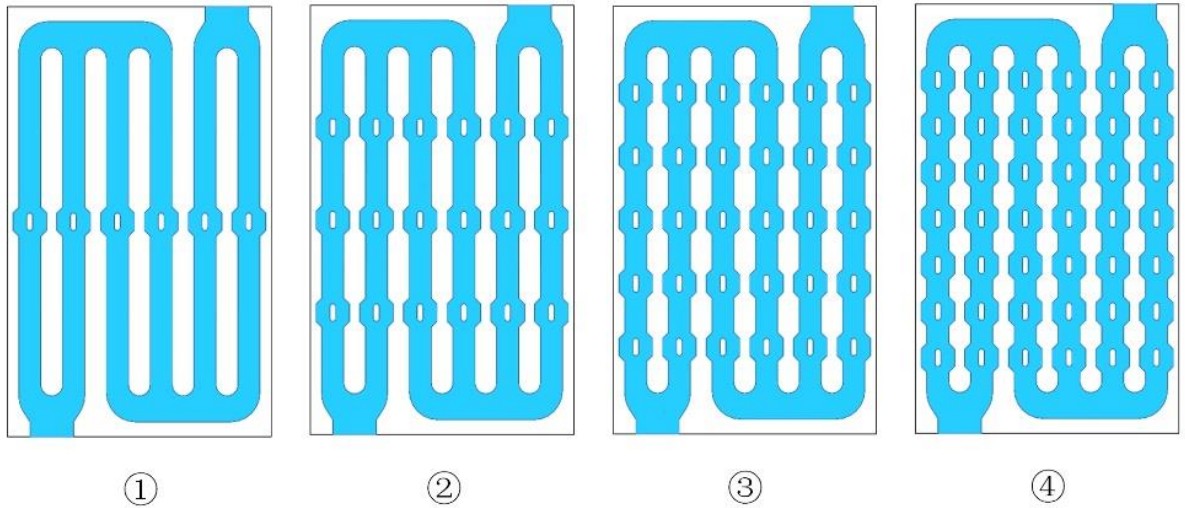

**Figure 6.** Different numbers of disturbance structures.

The average temperature distribution of the 12 batteries in the module at a 3 C discharge rate is shown in Figure 7. The average temperature of each battery decreases after disturbance structures are added to the mini-channel, and the decreasing trend increases when the disturbance structure number increases. When there are seven disturbance structures, the highest battery temperature is 36.18 °C, which is 0.63 °C lower than the initial plan, and the most obvious improvement in heat dissipation is achieved at this time. In addition, the addition of disturbance structures obviously changes the average temperature distribution of the batteries in the module. In the initial plan (no disturbance structure), the highest temperature occurs at the #4 battery, and the module temperature distribution shows the pattern of the temperature of the front batteries being higher than that of the back batteries. However, the temperature difference trend decreases a little when disturbance structures are added, and the temperature of all batteries decreases obviously when the number increases.

In addition to the average temperature of each battery, the maximum temperature of each battery is also monitored in the simulation. Figure 8 shows the maximum temperature and temperature difference of the battery module under different numbers of disturbances. With the addition of the disturbance structure, the maximum temperature and temperature difference both show a significant decrease compared to the initial plan. The lowest value appears when there are seven disturbance structures: the maximum temperature is 39.51 °C and the maximum temperature difference is 11.38 °C. Compared to the initial plan, the two values decrease by 0.75 °C and 0.41 °C, respectively. The phenomenon indicates that the flow field inside the mini-channel changes and the heat transfer is enhanced obviously,

which further improves the temperature rise and uniformity. Figure 9 shows the effect of different numbers of disturbance structures on the performance of the flow channel. It can be seen that disturbance structures increase the pressure drop and friction factor. The increase is small when the number of the disturbance structures increases from zero to five, but when the number of disturbance structures increases from five to seven, the pressure drop and friction factor are 7445.76 Pa and 1.25, which increase 14% and 31%, respectively, which indicates that the energy consumption is obvious. Considering the heat dissipation performance and energy consumption, plan 3 (five disturbance structures in the mini-channel) is adopted in the following analysis.

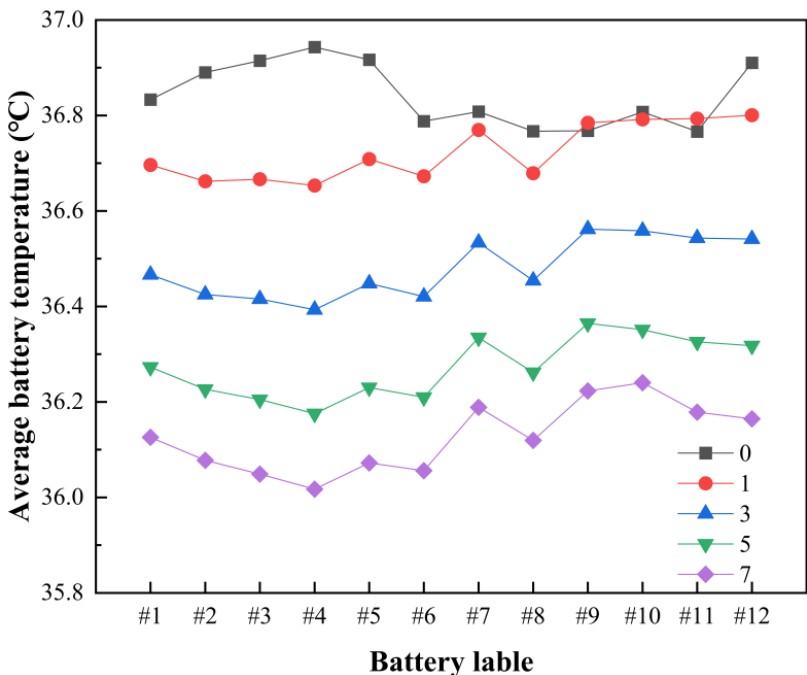

**Figure 7.** Battery temperature under different numbers of disturbance structures.

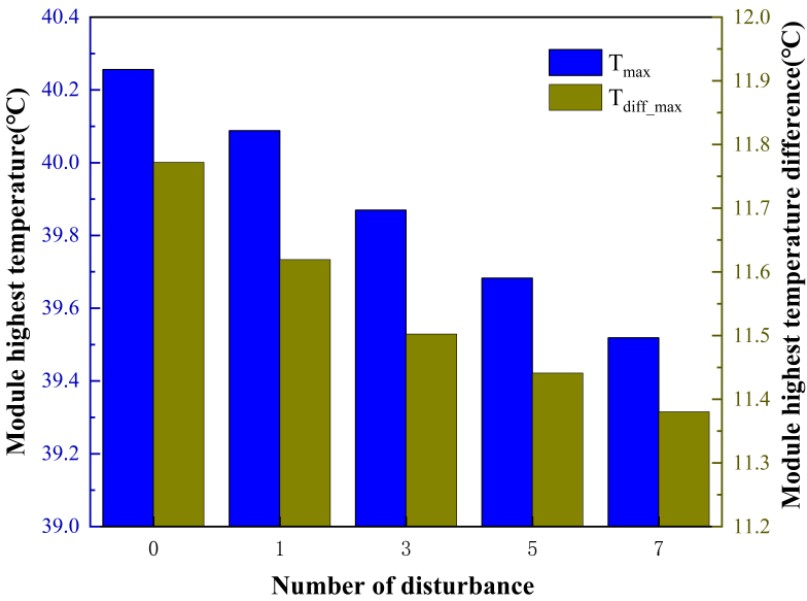

**Figure 8.** Module temperature under different numbers of disturbance structures.

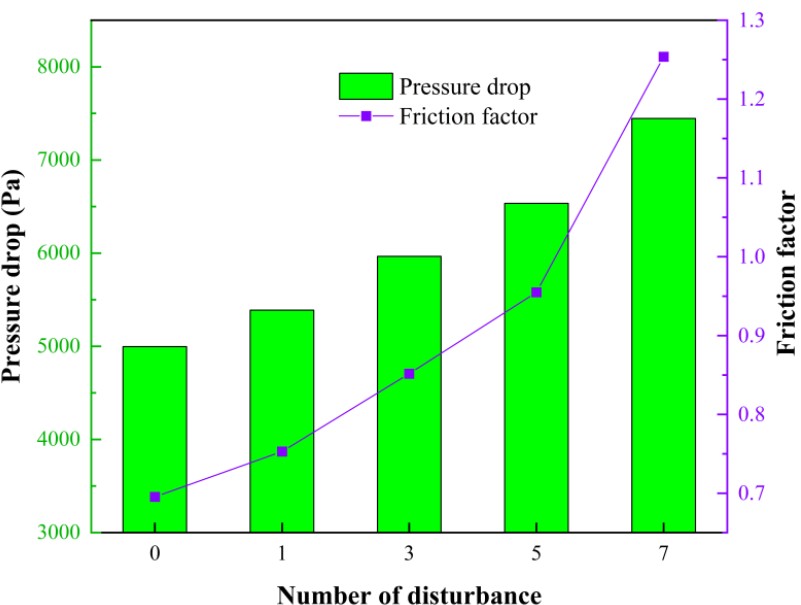

**Figure 9.** Channel performance under different numbers of disturbance structures.

### 3.2. Influence of Disturbance Structure Layout

Different distributions of the disturbance structures within the flow channel also affect the flow field of the coolant, which may improve the heat dissipation and the flow channel performance. Based on plan 3, four different distribution plans, numbered 5 to 8, are designed as shown in Figure 10. Plan 5 means the uniform distribution, plan 6 means the interlaced distribution, plan 7 means the gradually thinning distribution, and plan 8 means the gradually denser distribution. The structure parameters of the disturbance structure are the same as the initial value, and the simulation condition is still under the 3 C discharging rate.

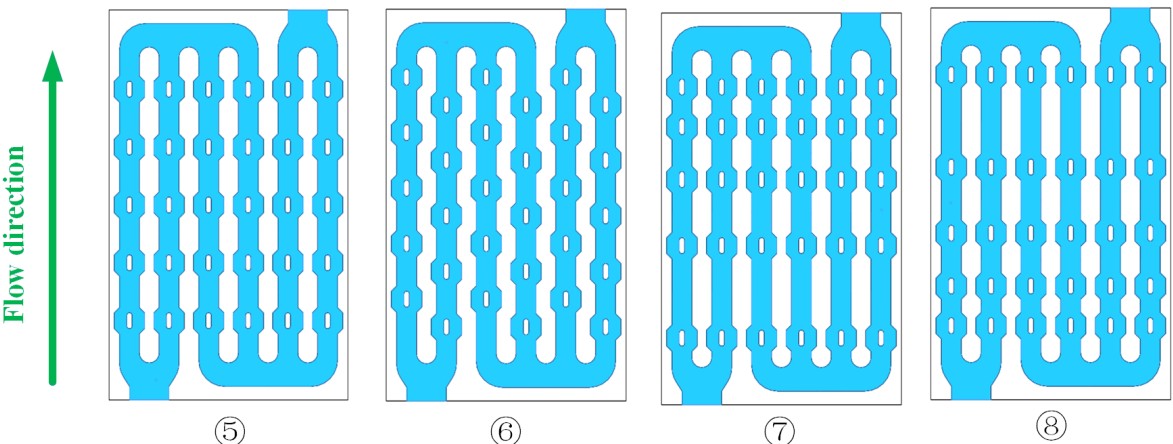

**Figure 10.** Different distributions of disturbance structures.

The effect of different distributions of the disturbance structures in the mini-channel on the average temperature of each battery is shown in Figure 11. The average temperature distribution of each battery is basically the same in the four plans, in which the average temperature of each battery is almost the lowest in plan 5 and is the highest in plan 8, which indicates the best and worst heat dissipation performance. In plan 5, the average temperature of battery #7 is the highest at 36.33 °C, and that of battery #4 is the lowest at 36.17 °C. The maximum average temperature difference is 0.16 °C, which indicates that the thermal balance performance is significant.

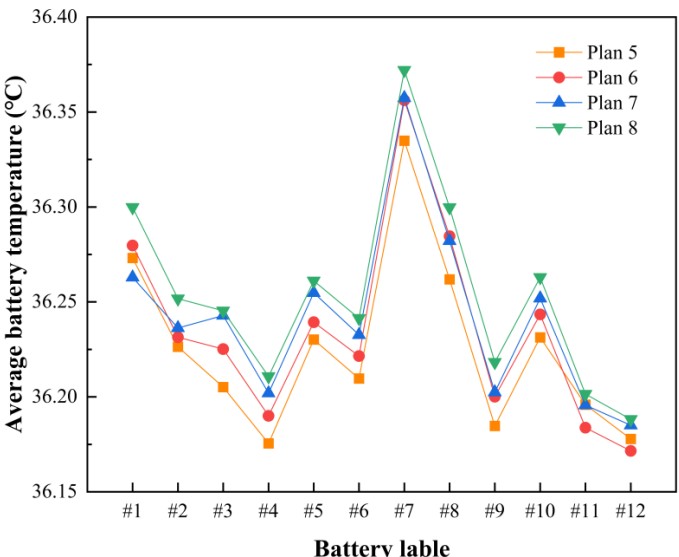

**Figure 11.** Average cell temperature under different distributions.

Figure 12 shows the maximum temperature and the maximum temperature difference under different distributions of disturbance structures. From the figure, it can be seen that the maximum temperature of the battery module in plan 5 is 36.68 °C, and the maximum temperature difference of the module in the remaining three plans is smaller but significantly higher than that of plan 5 (11.44 °C), which indicates that plan 5 has a greater advantage in temperature control. Figure 13 shows the influences of different distribution plans on the channel performance. The pressure drop and friction factor inside the mini-channel of plan 6 are the largest among the four plans, reaching 6743.98 Pa and 0.98, respectively, which indicates that the flow resistance is large and the operating power consumption of the pump is high. The difference in pressure drop between plan 5 and plan 7 is small; both are significantly lower than the remaining two plans, and while comparing the friction factors of the two plans, it can be seen that the friction factor of the mini-channel in the form of uniform distribution is 0.95, which is a larger decrease compared to plan 7. Therefore, the best performance of enhanced heat transfer within the mini-channel is achieved when five pieces of disturbance structures are uniformly distributed in each flow channel, and the following analysis of disturbance structure parameter optimization is carried out on this basis.

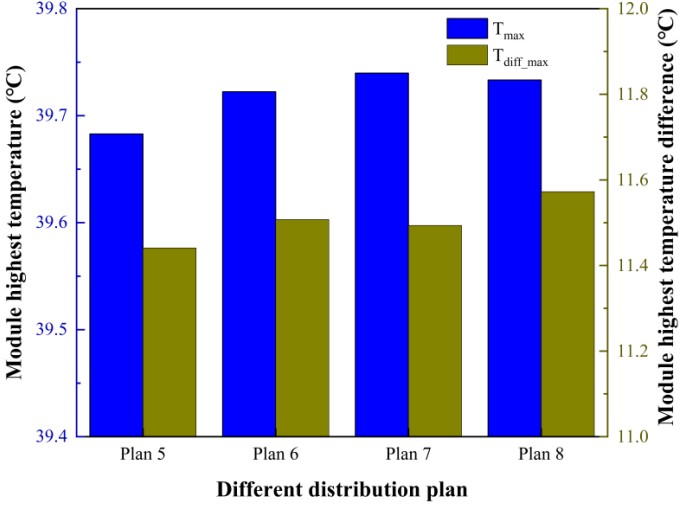

**Figure 12.** Module temperature under different distribution plans.

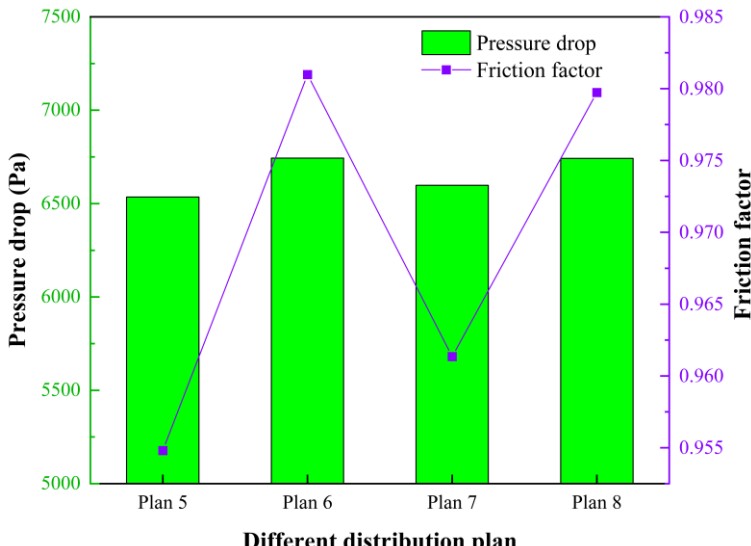

**Figure 13.** Channel performance under different distribution plans.

The coolant flow traces inside the mini-channel in the form of uniform distribution in plan 5 are shown in Figure 14. The coolant flow rate is relatively evenly distributed, with an average flow rate of about 0.8 m/s. As can be seen from the figure, the existence of the disturbance structure changes the original uniform flow field distribution. The flow velocity is relatively stable before the disturbance structure. However, after flowing through the disturbance structure, the growth of the fluid boundary layer is promoted, a new boundary layer is continuously generated in the middle region of the fluid, and even low-speed secondary flow and reflux will appear in the local region; the coolant is continuously disturbed and guided to the cavities on both sides, and the flow state of the fluid is abruptly changed so that its flow is constantly close to the wall of the flow channel, which increases the turbulence and accordingly enhances convective heat transfer in the mini-channel.

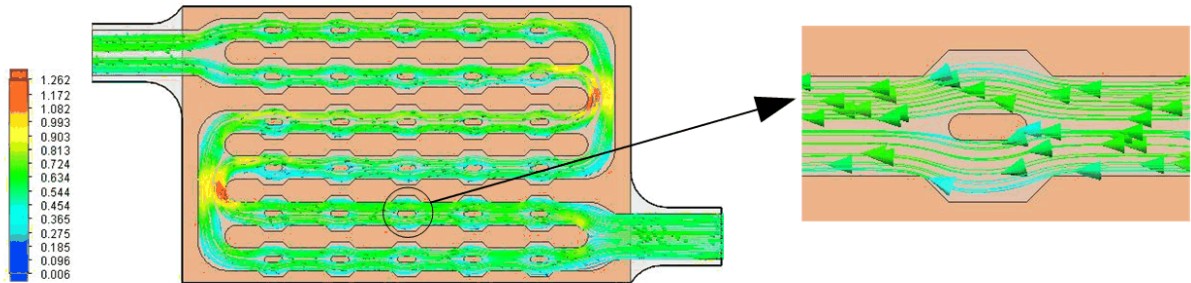

**Figure 14.** Fluid trace inside mini-channel in plan 5.

Figure 15 shows the change in the velocity vector of the coolant before and after flowing through the disturbance structure. When approaching the disturbance structure, the middle region of the fluid already starts to flow to both sides of the flow channel and creates a new boundary layer in the middle region. When reaching the middle region of the disturbance structure, the flow of fluid to both sides becomes more intense, and there is even backflow and secondary flow on both sides of the wall of the disturbed fluid. After passing through the disturbance structure, the fluids on both sides converge to the middle region again, and the boundary layer in the middle region gradually disappears. The process repeats at each disturbance structure, which promotes heat exchange with the wall during the lateral flow, thus enhancing the heat dissipation performance.

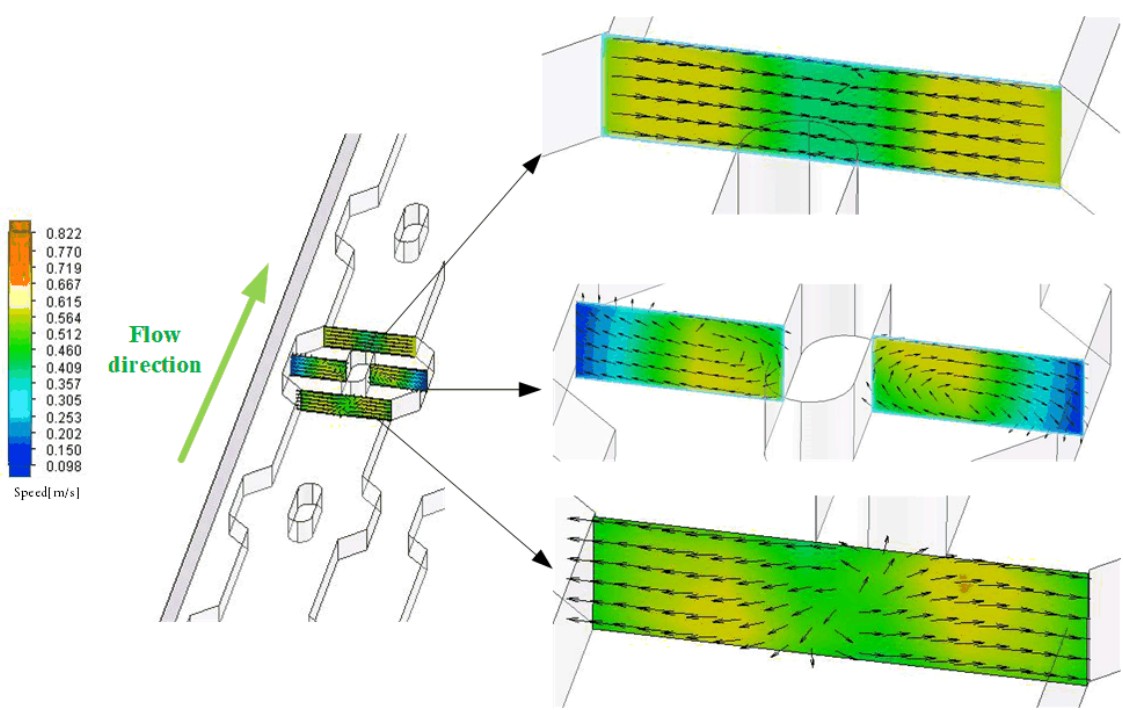

**Figure 15.** Cross-sectional view of flow velocity at different locations.

## 4. Parameter Optimization of the Disturbance Structure

### 4.1. Orthogonal Experiment Design

Changing the parameters of the disturbance structure will directly affect the heat dissipation performance of the mini-channel liquid cooling plate, and the processes of reconstructing the finite element model and conducting numerical simulation will be repeated many times if the parameters are changed one by one. This way may be overly demanding in terms of computation resources and time. In order to improve computational efficiency, the orthogonal test method is used to design the parameter optimization process [30]. Based on plan 5, the design parameters of the disturbance structure include space between adjacent disturbance structures d1, length d2, width d3, and tilt angle $\beta$. The factors and levels for this orthogonal experiment are shown in Table 3.

**Table 3.** Experimental factors and levels.

| Level | Factor | | | |
|---|---|---|---|---|
| | **d1 (mm)** | **d2 (mm)** | **d3 (mm)** | **$\beta$ (°)** |
| L1 | 14 | 3 | 1.5 | 15 |
| L2 | 16 | 4 | 2 | 30 |
| L3 | 18 | 5 | 2.5 | 45 |
| L4 | 20 | 6 | 3 | 60 |

According to the test factors and levels in Table 3, the $L_{16}$ ($4^4$) orthogonal test table was designed to carry out the orthogonal analysis. Adjacent disturbance structures d1, length d2, width d3, and tilt angle $\beta$ were selected as the factors. The maximum temperature $T_{\max}$, maximum temperature difference $T_{\mathrm{diff}}$, and friction factor $f$ were selected as the evaluation indexes of the heat dissipation performance. Numerical simulations were conducted for each scheme under the same conditions: 3 C discharging rate, 25 °C the coolant inlet temperature, and 400 L/h coolant flow rate. The detailed scheme of the orthogonal experiment and results are shown in Table 4.

**Table 4.** Orthogonal test and simulation results.

| Number | Factor | | | | Evaluation Index | | |
|---|---|---|---|---|---|---|---|
| | d1 (mm) | d2 (mm) | d3 (mm) | $\beta$ (°) | $T_{max}$ (°C) | $T_{diff}$ (°C) | $f$ |
| 1 | L1 (14) | L1 (3) | L1 (1.5) | L1 (15) | 39.8433 | 11.4737 | 1.0430 |
| 2 | L1 (14) | L2 (4) | L2 (2) | L2 (30) | 39.8168 | 11.5788 | 1.1270 |
| 3 | L1 (14) | L3 (5) | L3 (2.5) | L3 (45) | 39.7650 | 11.5139 | 1.0949 |
| 4 | L1 (14) | L4 (6) | L4 (3) | L4 (60) | 39.7281 | 11.5574 | 0.9870 |
| 5 | L2 (16) | L1 (3) | L2 (2) | L3 (45) | 39.8034 | 11.4672 | 1.0533 |
| 6 | L2 (16) | L2 (4) | L1 (1.5) | L4 (60) | 39.7154 | 11.4105 | 0.9154 |
| 7 | L2 (16) | L3 (5) | L4 (3) | L1 (15) | 39.6499 | 11.5349 | 1.6825 |
| 8 | L2 (16) | L4 (6) | L3 (2.5) | L2 (30) | 39.7101 | 11.5225 | 1.2705 |
| 9 | L3 (18) | L1 (3) | L3 (2.5) | L4 (60) | 39.6481 | 11.4087 | 0.9787 |
| 10 | L3 (18) | L2 (4) | L4 (3) | L3 (45) | 39.7109 | 11.4318 | 1.1623 |
| 11 | L3 (18) | L3 (5) | L1 (1.5) | L2 (30) | 39.7883 | 11.4652 | 1.0405 |
| 12 | L3 (18) | L4 (6) | L2 (2) | L1 (15) | 39.7728 | 11.4790 | 1.2455 |
| 13 | L4 (20) | L1 (3) | L4 (3) | L2 (30) | 39.6904 | 11.4237 | 1.4371 |
| 14 | L4 (20) | L2 (4) | L3 (2.5) | L1 (15) | 39.7117 | 11.4396 | 1.3585 |
| 15 | L4 (20) | L3 (5) | L2 (2) | L4 (60) | 39.6155 | 11.3776 | 0.9546 |
| 16 | L4 (20) | L4 (6) | L1 (1.5) | L3 (45) | 39.6868 | 11.4347 | 0.9958 |

*4.2. Range Analysis of Orthogonal Experiment Results*

The range analysis method includes two steps of calculation and judgment. From the orthogonal test simulation results in Table 4, the sum of the factors $K_i$ in each column can be calculated, and thus the average value $k_i$ of $K_i$ can be calculated as well. Based on the average value $k_i$, the range value of each factor for the evaluation indexes can be obtained according to Equation (15).

$$R_j = \max\{k_1, k_2, k_3, k_4\} - \min\{k_1, k_2, k_3, k_4\} \tag{15}$$

Therefore, the sensitivity of each structural parameter of the disturbance structure to different evaluation indexes can be visualized by the range value $R$ derived from the orthogonal experiment. A larger $R$ value indicates a greater sensitivity, i.e., a greater influence of the factor on the evaluation index. The results of the range analysis are shown in Table 5.

According to the results in Table 5, the sensitivity of each structural parameter of the disturbance structure is in the order of $d_1 > \beta > d_3 > d_2$ for the evaluation index maximum temperature, and the detailed values are $d_1$ = 20 mm, $d_2$ = 5 mm, $d_3$ = 3 mm, and $\beta$ = 60°; for the maximum temperature difference, the order is $d_1 > \beta > d_2 > d_3$, and the detailed values are $d_1$ = 20 mm, $d_2$ = 3 mm, $d_3$ = 1.5 mm, and $\beta$ = 60°; for the friction factor, the order is $\beta > d_3 > d_1 > d_2$, and the detailed values are $d_1$ = 14 mm, $d_2$ = 6 mm, $d_3$ = 1.5 mm, and $\beta$ = 60°.

With $d_1$ as the horizontal coordinate and $k$ as the vertical coordinate, the trend of the influence of the disturbance structure spacing on each evaluation index can be obtained, as shown in Figure 16. The maximum temperature difference of the battery module gradually decreases as the spacing of the disturbance structure increases, because the larger spacing indicates that the disturbance structure is more uniformly distributed inside the mini-channel, and the plate can evenly dissipate the heat to improve the temperature uniformity. In addition, the maximum temperature and the friction factor show an opposite trend, which indicates that the two indexes are negatively correlated. In this section, the

maximum temperature and the maximum temperature difference achieve the best value when $d_1 = 20$ mm, and this value is selected as the final parameter, although the friction factor is not optimal. Here, the heat dissipation performance is considered first.

**Table 5.** Range analysis results.

| Index | Parameter | d1 (mm) | d2 (mm) | d3 (mm) | β (°) |
|---|---|---|---|---|---|
| | | | **Factor** | | |
| $T_{max}$ | $k_1$ | 39.7883 | 39.7463 | 39.7585 | 39.7444 |
| | $k_2$ | 39.7197 | 39.7387 | 39.7521 | 39.7514 |
| | $k_3$ | 39.7323 | 39.7047 | 39.7101 | 39.7415 |
| | $k_4$ | 39.6761 | 39.7245 | 39.6948 | 39.6768 |
| | R | 0.1122 | 0.0416 | 0.0637 | 0.0746 |
| | Sensitivity | | $d_1 > \beta > d_3 > d_2$ | | |
| | Best solution | | $d_1 = 20$ mm, $d_2 = 5$ mm, $d_3 = 3$ mm, $\beta = 60°$ | | |
| $T_{diff}$ | $k_1$ | 11.5310 | 11.4433 | 11.4427 | 11.4818 |
| | $k_2$ | 11.4838 | 11.4652 | 11.4757 | 11.4976 |
| | $k_3$ | 11.4462 | 11.4729 | 11.4712 | 11.4619 |
| | $k_4$ | 11.4189 | 11.4984 | 11.4878 | 11.4386 |
| | R | 0.1121 | 0.0551 | 0.0451 | 0.059 |
| | Sensitivity | | $d_1 > \beta > d_2 > d_3$ | | |
| | Best solution | | $d_1 = 20$ mm, $d_2 = 3$ mm, $d_3 = 1.5$ mm, $\beta = 60°$ | | |
| $f$ | $k_1$ | 1.0630 | 1.1280 | 0.9987 | 1.3324 |
| | $k_2$ | 1.2304 | 1.1408 | 1.0951 | 1.2188 |
| | $k_3$ | 1.1068 | 1.1931 | 1.2323 | 1.0766 |
| | $k_4$ | 1.1865 | 1.1247 | 1.3172 | 0.9589 |
| | R | 0.1674 | 0.0684 | 0.3185 | 0.3735 |
| | Sensitivity | | $\beta > d_3 > d_1 > d_2$ | | |
| | Best solution | | $d_1 = 14$ mm, $d_2 = 6$ mm, $d_3 = 1.5$ mm, $\beta = 60°$ | | |

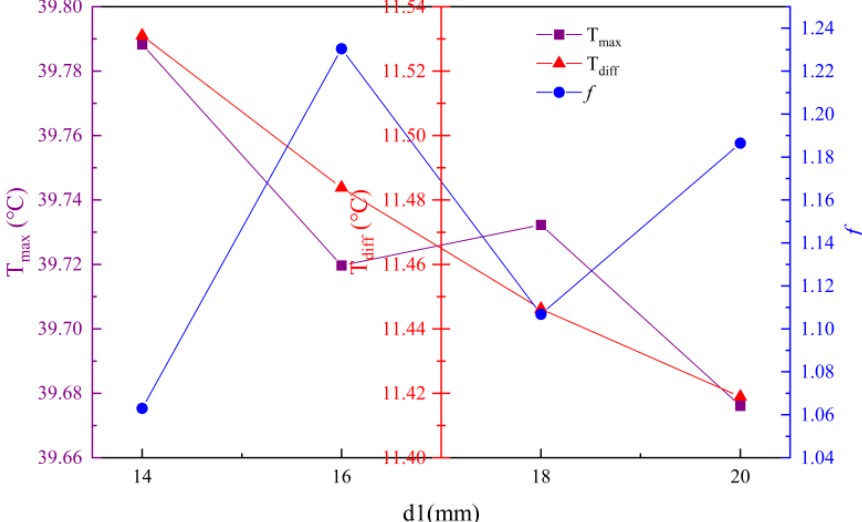

**Figure 16.** Influence of $d_1$ on evaluation indexes.

With $d_2$ as the horizontal coordinate and $k$ as the vertical coordinate, the trend of the influence of the disturbance structure length on each evaluation index can be obtained, as shown in Figure 17. As $d_2$ increases, the maximum temperature difference gradually increases, which indicates that increasing the length is not conducive to temperature

uniformity. Actually, increasing $d_2$ means the distance between adjacent disturbance structures decreases, which is not conducive to the uniform distribution of the fluid, and the heat dissipation is better in the area where the disturbance structure exists. In addition, the maximum temperature and the friction factor show an opposite trend. On the one hand, increasing the length will cause the area of convective heat transfer inside the mini-channel to increase, thus improving the heat dissipation performance; on the other hand, increasing the length will disturb the original flow traces of the coolant, the velocity and pressure fields inside the mini-channel will change, and the coolant will diverge when it is close to the disturbance structure, so the frictional resistance inside the flow channel will also increase. However, when the length reaches 5 mm, increasing the length will cause the maximum temperature to rise, which indicates that when the disturbance structure is too long, the spacing will become too short, and the coolant will only flow from both sides of the disturbance structure. There will be a dead zone between the adjacent disturbance structures, and the friction factor will become smaller, but at the same time, the heat dissipation performance will be affected, and when the length is 5 mm, the enhanced heat transfer performance of the mini-channel liquid cooling plate is the best.

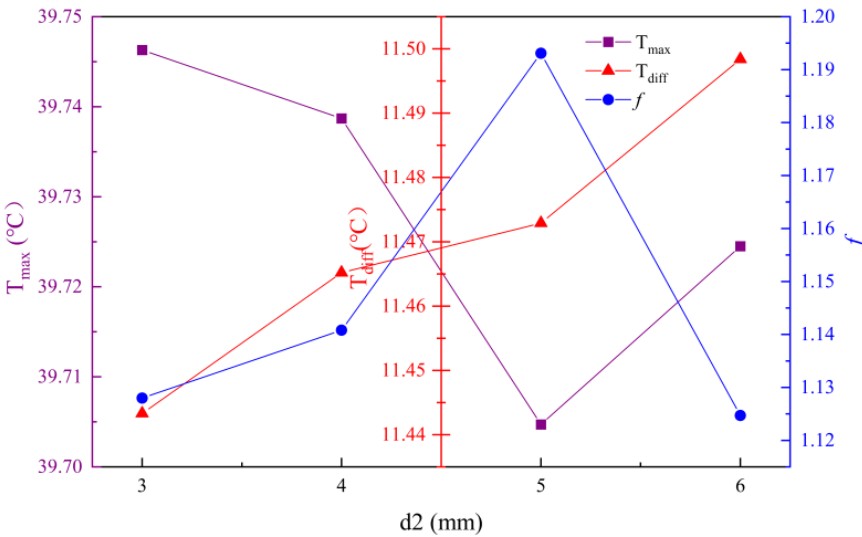

**Figure 17.** Influence of $d_2$ on evaluation indexes.

With $d_3$ as the horizontal coordinate and $k$ as the vertical coordinate, the trend of the influence of the disturbance structure width on each evaluation index can be obtained, as shown in Figure 18. As $d_3$ increases, the maximum temperature decreases, but the friction factor gradually increases. On the one hand, increasing the width of the disturbance structure causes an increase in the convective heat transfer area inside the mini-channel; on the other hand, the disturbing effect on the fluid is enhanced, the stable flow field is abruptly changed, the boundary layer is generated in the middle region of the fluid, and the turbulence is enhanced. In addition, the overall trend of the maximum temperature difference is increasing, which indicates that increasing the width will cause the coolant to divert to both sides of the disturbance structure, and the coolant flow rate is lower in the area between adjacent disturbance structures, resulting in uneven distribution of coolant flow and preventing the battery surface from being uniformly dissipated, thus causing poor temperature uniformity.

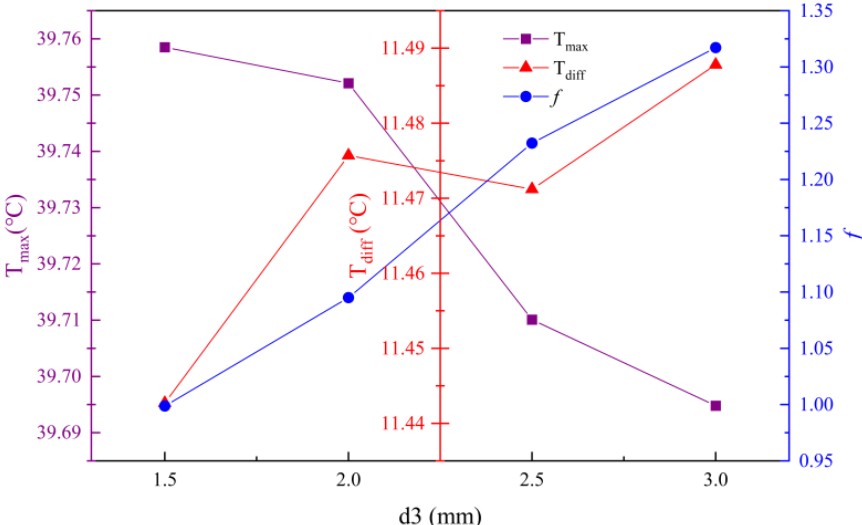

**Figure 18.** Influence of $d_3$ on evaluation indexes.

With $\beta$ as the horizontal coordinate and $k$ as the vertical coordinate, the trend of the influence of the disturbance structure width on each evaluation index can be obtained, as shown in Figure 19. The friction factor gradually decreases as $\beta$ increases, which indicates that the coolant in the cavity is subject to less frictional resistance when flowing out of the cavity, which is more conducive to the natural flow of the fluid. However, the maximum temperature and the maximum temperature difference increase slightly and then decrease significantly when $\beta$ increases to 30°, which indicates that the heat dissipation is the worst at this value. At this value, the coolant flow velocity is low at the opening angle of the cavity, and there is even a dead zone, so the coolant cannot flow evenly to dissipate heat. In addition, as shown in Table 5, the sensitivity of the friction factor to each evaluation index is the lowest among all factors, which indicates that it has the least influence on the enhanced heat transfer performance, and the thermal characteristics of the battery module and the performance of the flow channel are the best when the cavity opening angle is 60°.

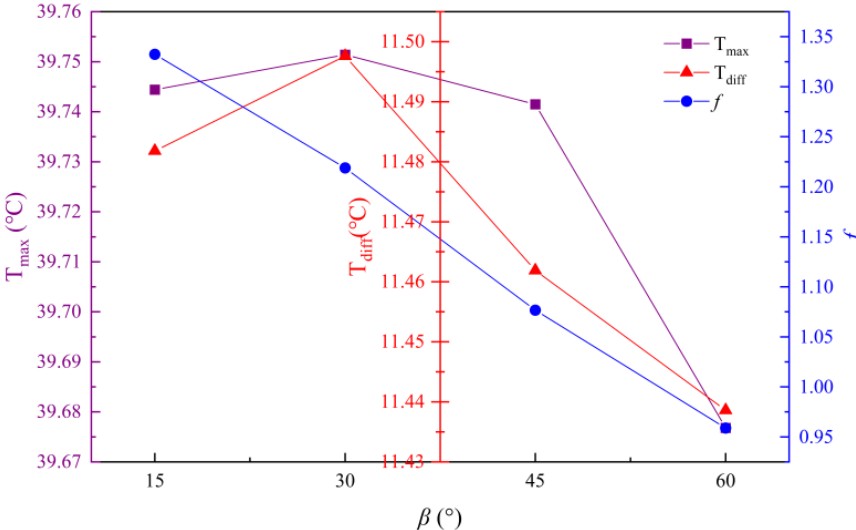

**Figure 19.** Influence of $\beta$ on evaluation indexes.

Based on the determined parameters, a comparison between the BTMS in this paper and a BTMS with passive air cooling is performed. In the air-cooling system, the module is the same as that in this paper without cooling plates and thermal conductive sheets. All batteries are in contact with each other. The environmental conditions and discharging

rate are the same. In addition, the computation domain is 20 mm larger than each side of the module, considering that the battery pack is compact in real-world applications. The results are shown in Table 6. It can be seen that the maximum average temperature cannot be controlled with a compact module structure under passive air cooling, and it reaches 49.06 °C. The maximum average temperature difference reaches 6.41 °C, 5.48 °C higher than before.

**Table 6.** Comparison with passive air cooling.

| Cooling Type | Mini-Channel | Passive Air Cooling |
| --- | --- | --- |
| Max average temperature (°C) | 35.83 | 49.06 |
| Max average temperature difference (°C) | 0.93 | 6.41 |
| Discharging rate | 3 C | 3 C |

## 5. Conclusions

An efficient battery thermal management system plays an important role in electric vehicle operation. In this paper, a novel liquid-cooling system based on mini-channel plates with disturbance structures is proposed. In order to design the cooling system, the battery model is established first. Verified by heat generation experiments, the model achieves high accuracy with an error of less than 4%. Then, the battery module model consisting of 12 batteries and 6 cooling plates is established. Five plate designs are proposed first with zero, one, three, five, and seven disturbance structures, respectively. Plan 3 (five disturbance structures) is determined by considering the heat dissipation performance and flow channel performance. Then, four layout plans of the disturbance structures are proposed. Results show that plan 5 (disturbance structures distributed evenly) achieves the best performance both in the heat dissipation performance and flow channel performance. Under a 3 C discharging rate, the highest average temperature is 36.33 °C and the maximum average temperature difference is 0.16 °C. Based on plan 5, the orthogonal experiment and range analysis are adopted especially for the optimization of the disturbance structures. The factors include the space between adjacent disturbance structures d1, length d2, width d3, and tilt angle $\beta$. The evaluation indexes include the maximum temperature, maximum temperature difference, and friction factor. Results of the range analysis show that the best combination of the four parameters is d1 = 20 mm, d2 = 5 mm, d3 = 1.5 mm, and $\beta = 60°$.

**Author Contributions:** Methodology, study design, experiment, model, R.L.; software, data analysis, data collection, Y.Y.; writing, literature search, figures, F.L.; data collection, data analysis, writing modification, J.L.; study design, results discussion, methodology modification, X.C.; funding acquisition, J.L. and X.C. All authors have read and agreed to the published version of the manuscript.

**Funding:** The project is supported in part by the Prospective Study Funding of Nanchang Automotive Innovation Institute, Tongji University (No. QZKT2020-01), National Natural Science Foundation of China (No. 62103415).

**Data Availability Statement:** All the research data has been provided in this paper.

**Conflicts of Interest:** The authors declare no conflict of interest.

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
