# Peer review of "Investigation on Battery Thermal Management Based on Enhanced Heat Transfer Disturbance Structure within Mini-Channel Liquid Cooling Plate"

_electronics, doi:10.3390/electronics12040832_

Round 1

Reviewer 1 Report

The authors have created a comprehensive work and provide balanced information on the theoretical basis, mathematical processing, simulations and experimental verification of a new model of the mini-channel cooling plate for maintaining the temperature of a battery system within normal limits.

Overall, this work is  interesting, even though some minor issues should be addressed before being published on “Energy.”

This article can be improved by addressing the following issues:

1. Do you think you should specify more information about the cooling plate, i.e. how it was made and from what material?

2. cooling water inlet paths to the six plates as they were made?

3. a performance comparison with a passive cooling system made with air.

Reviewer 2 Report

1.       Meshing details are to be given such as element type, inflation etc.

2.       Grid dependency test  has to be done

3.       How to you justify the increased pressure drop by increasing the number of disturbance points in one row?

4.       In figure 10 why battery 7 shows a very high temperature compared to others?
